# Effects of Abscisic Acid on the Physiological and Biochemical Responses of *Saccharina japonica* Under High-Temperature Stress

**DOI:** 10.3390/ijms252111581

**Published:** 2024-10-28

**Authors:** Jiexin Cui, Yinru Dai, Yichang Lai, Yenzhen Tan, Tao Liu

**Affiliations:** State Key Laboratory of Marine Environmental Science, College of Ocean and Earth Sciences, Xiamen University, Xiamen 361102, China; cuijiexin@stu.xmu.edu.cn (J.C.);

**Keywords:** abscisic acid, *Saccharina japonica*, high-temperature stress, gene expression regulation

## Abstract

*Saccharina japonica* is one of the most productive aquatic plants in the world, widely used in food, feed, medicine, and other industries. Predominantly inhabiting temperate marine environments in mid- to high-latitude regions of the Northern Hemisphere, the growth of *S. japonica* is significantly limited by high-temperature stress. Abscisic acid (ABA) plays an important role in plant growth and development and stress responses. However, the role of ABA on high-temperature stress tolerance in *S. japonica* still needs to be further elucidated. Here, we found that exogenous ABA significantly alleviated disease and decay in *S. japonica* under high-temperature stress while also increasing the relative growth rate, chlorophyll fluorescence parameters, photosynthetic pigment, and osmotic substance content. Meanwhile, exogenous ABA enhanced the activity of protective enzymes and up-regulated the transcript levels of antioxidant-related genes, thereby reducing oxidative damage. Most importantly, we observed a significant increase in ABA content and the transcript levels of key genes involved in ABA synthesis in *S. japonica* under high-temperature stress, which were further amplified by the addition of exogenous ABA. In conclusion, this study provides evidence that ABA can moderate the detrimental effects of high-temperature stress and provides a theoretical basis for the screening of *S. japonica* germplasm resources and the cultivation of new stress-resistant varieties.

## 1. Introduction

*Saccharina japonica* is a significant economic brown alga belonging to the Ochrophyta, Phaeophyceae, Laminariales, and Laminariaceae [1]. *S. japonica* exhibits a heteromorphic alternation of generations, comprising the sporophyte generation of multicellular thallus and gametophyte generation of unicellular and uniseriate cell filaments, making it valuable for studying the evolution of multicellular algae. As the principal species in worldwide aquaculture, with an annual production of 10,861 kilotons, accounting for 29.75% of global aquatic plant aquaculture (FAO, 2022), it is extensively utilized across various sectors, including food, medicine, feed, and other industries [2,3]. In recent years, scientific research has demonstrated that kelp farming has the potential to absorb significant quantities of carbon dioxide, thereby creating a carbon sink effect that can contribute to the mitigation of ocean acidification resulting from global climate change [4,5]. *S. japonica* is a temperate marine species mainly found in the mid- and high-latitude zones of the Northern Hemisphere [6]. Therefore, high-temperature stress has become one of the major abiotic stresses that severely constrain *S. japonica* growth, which can lead to excessive light energy absorption, decreased photosynthetic efficiency, a reduced CO_2_ assimilation rate, and significantly lower yields [7].

Phytohormones are organic signaling molecules produced by plants through their own metabolism that can produce significant physiological effects at very low concentrations [8]. Abscisic acid (ABA) plays an important role in plants [9] by not only contributing to seed maturation and dormancy but also playing a crucial role in various stress environments (cold, heat, high salt, drought stress, etc.) [10,11,12], known as the adversity resistance hormone [13,14]. It has been demonstrated that ABA plays a crucial role in responding to adversity and is mainly involved in plant responses to high-temperature stress by mediating their antioxidant capacity, heat shock proteins (HSPs), and sugar metabolism [14,15]. ABA is also present in algae and is usually present in lower levels than in higher plants, but it also plays a more important role in the physiological activities of algae [16,17]. ABA has been identified in about 100 species of algae in the Rhodophyta, Ochrophyta, Cryptophyta, Bacillariophyta, Euglenophyta, and Chlorophyta [18], with levels varying among different algae, ranging from about 0.9 to 34 nmol·kg^−1^ [19,20,21]. In aquatic plants, ABA has also shown strong resistance to stress [22]. ABA can help *Myriophyllum aquaticum* adapt to cold [23] and reduce the dehydration rate of *Pyropia haitanensis* by promoting cell mechanisms that lower evaporation rates [24]. The exogenous addition of ABA could promote the growth of *Gracilariopsis lemaneiformis* under high-temperature stress and increase the content of mannitol, proline, and alginate [25]. Moreover, exogenous ABA can enhance the absorption of HCO_3_^−^ by *Chlamydomonas reinhardtii* under strong light and mitigate dehydration-induced damage by inhibiting reactive oxygen species production and enhancing the expression of antioxidant enzymes [26,27,28].

At present, the research on ABA in *S. japonica* mainly focuses on content detection, and relatively few studies on the regulation of its expression and mechanism of action under adversity have been completed. The ABA content in *S. japonica* ranged from 2.5 to 3.06 μg·g^−1^ [21], varying across growth periods and peaking at maturity, with the highest levels observed at the base [29]. Previous studies have demonstrated that high-temperature stress induces alterations in antioxidant enzymes, photosynthesis, and osmoregulatory substances in *S. japonica* sporophytes [30]. This phenomenon is also observed in *S. japonica* gametophytes, though the consequences are typically more severe and often lethal [31,32]. However, there is no report to elucidate the changes in ABA in *S. japonica* under high-temperature stress, and the physiological and biochemical changes in *S. japonica* after the addition of exogenous ABA under high-temperature stress are also unknown.

In this study, we first investigated the effects of high-temperature stress on the physiology and biochemistry of *S. japonica*, including the determination of ABA content and the expression level of key genes of the synthesis pathway, the determination of the content and activity of antioxidant enzymes such as superoxide dismutase (SOD), peroxidase (POD), and catalase (CAT), and the study of the transcription levels of related genes. We also assayed photosynthesis-related substances such as chlorophyll (Chl), the maximum quantum efficiency of photosystem II (Fv/Fm), and carotenoids and measured the content of osmotic regulators, such as proline (Pro). Meanwhile, the effects of exogenous ABA on the physiological and biochemical properties of *S. japonica* were further investigated from the above aspects in order to better understand the mechanism of endogenous ABA in *S. japonica* and the effects of exogenous ABA on the synthesis and metabolism of antioxidant enzymes and the regulation of gene expression.

## 2. Results

### 2.1. Effect of Abscisic Acid on Physiological Morphology and Characteristics of S. japonica Under High-Temperature Stress

High-temperature stress caused significant damage to *S. japonica*, with the severity of the damage increasing at higher temperatures. At 21 °C, large rotting spots and missing tails were observed on the middle and lower parts of the thallus, but with the exogenous addition of ABA, only a few rotting spots were present, and the algal morphology remained mostly intact (Figure 1). The relative growth rate (RGR) increased by 20.31% under control conditions (12 °C) and 9.78% under the condition of 18 °C. RGR was negative at 21 °C, indicating a decline in the growth of *S. japonica*, and RGR increased by 10.89% under 21 °C +ABA treatment was greater than under the condition of 18 °C (Figure 2a). The survival rate followed a similar trend, with the lowest rate 60%, observed in the 21 °C treatment group (Figure 2b).

### 2.2. Effect of Abscisic Acid on Photosynthetic Characteristics of S. japonica Under High-Temperature Stress

As shown in Figure 3a, the control group (CK) maintained a relatively stable Fv/Fm, with only minor fluctuations over time. In contrast, the Fv/Fm of the 21 °C treatment group showed a decreasing trend with prolonged stress time and reached its minimum at 72 h, a decrease of 72.2% compared to the initial conditions, showing significant differences. At 18 °C, the Fv/Fm reached the minimum at 6 h and then increased slowly, and under the condition of 21 °C +ABA, the Fv/Fm reached the minimum at 12 h. The Chl content of all treatment groups decreased with prolonged stress under a high temperature, showing significant differences in the Chl content at the later stages of cultivation under different high-temperature stresses, with the effect of ABA on the Chl content under high-temperature stress not significant (Figure 3b). The chlorophyll a (Chl a) content in the high-temperature stress treatment groups was lower than in the CK group, showing significant differences from the CK group after 24 h, and the Chl a content under the 21 °C +ABA condition was significantly different from that without ABA (Figure 3c). The content of chlorophyll b (Chl b) at 18 °C reached its lowest point at 24 h, decreasing by 52.55% compared to the CK. At 21 °C, the Chl b content fluctuated with a jagged change, and under the condition of 21 °C +ABA, it reached its lowest point at 48 h, with no significant difference compared to the without-ABA condition (Figure 3d).

Four carotenoids were detected in *S. japonica* under high-temperature stress, including zeaxanthin, violaxanthin, *α*-carotene, and *β*-carotene. The most abundant component in *S. japonica* was *β*-carotene, followed by violaxanthin. Zeaxanthin content remained stable at 18 °C but fluctuated at 21 °C and reached a maximum value of 0.1501 ng·g^−1^ Fw at 48 h, which was 6.9- and 9-fold higher compared to the CK and 18 °C groups, respectively. The zeaxanthin content under 21 °C +ABA increased with the extension of the culture period and reached the maximum value of 0.1533 ng·g^−1^ Fw at 72 h, which was significantly increased by 65.73% compared with that of the 21 °C treatment group (Figure 4a). This indicates that ABA had a significantly enhanced effect on the increase in zeaxanthin content in *S. japonica* under high-temperature stress. Compared with the CK group, the violaxanthin content decreased in almost all treatment groups, but in the 18 °C group, it was significantly higher than the CK group at 72 h, and ABA had no significant effect on the violaxanthin content (Figure 4b). There was no significant change in the *α*-carotene content in all treatment groups compared to the CK group (Figure 4c). The *β*-carotene content of the CK group did not change significantly during the culture period, and it fluctuated in each treatment group. Under 18 °C conditions, the highest content of *β*-carotene was 2.8931 ng·g^−1^ Fw, which increased by 52.83% and 70.64% compared to the CK group and the 21 °C group, respectively. The addition of ABA had no significant effect on the *β*-carotene content (Figure 4d).

### 2.3. Effect of Abscisic Acid on Physiological Indicators of S. japonica Under High Temperature Stress

As the duration of treatment increased, there was no significant change in SOD activity in the CK group; the treatment group first increased and then decreased. There was no significant difference between the 18 °C and 21 °C groups. Under 21 °C +ABA conditions, the SOD activity showed a significant difference compared with the no-addition variation, with the most significant difference at 48 h. The addition of ABA caused a 13% decrease in the SOD activity of *S. japonica* under high-temperature stress at 21 °C (Figure 5a). POD activity showed a general trend of increasing and then decreasing, as shown in Figure 5b; all treatment groups were higher than the CK group after 12 h. At 24 and 48 h, the POD activity increased by 19.95% and 16.87% at 21 °C compared with 18 °C, respectively. At 24 h, POD activity was significantly decreased by the addition of ABA under the condition of 21 °C.

The CAT activity in the treatment group increased with the cultivation time and was higher than in the CK group. There was no significant change in CAT activity in the CK group (Figure 5c). After 72 h of stress, the CAT activity of the treatment groups reached the maximum values of 63.91 U·g^−1^ Fw (18 °C), 67.55 U·g^−1^ Fw (21 °C), and 70.01 U·g^−1^ Fw (21 °C +ABA), respectively, and there was no significant difference between the different temperatures. The addition of ABA under 21 °C stress increased the CAT activity by 3.64% compared to the no-addition variation. The ascorbate peroxidase (APX) activity of the CK group showed no significant changes (Figure 5d). The 18 °C and 21 °C groups increased with the cultivation time, and the APX activity reached a maximum value of 2.235 U·g^−1^ Fw and 3.919 U·g^−1^ Fw at 72 h, which significantly increased the activity by 31.78% and 131.07%, respectively, compared to CK. Under 21 °C +ABA conditions, the APX activity first increased and then decreased and reached its maximum value at 24 h, with significant differences compared to without ABA after 12 h.

Malondialdehyde (MDA) content is an indicator of lipid peroxidation. The MDA content in the CK group showed a small upward and downward fluctuation (Figure 5e). The MDA content of the treatment group showed an increasing trend throughout the treatment. Among the high-temperature treatment groups, the difference in MDA content between the 18 °C and 21 °C groups were not significant. The MDA content in the 21 °C +ABA group was reduced by 21.99% and 32.77% compared to the 21 °C group at 6 h and 48 h, respectively. This indicates that the addition of exogenous ABA under high-temperature stress could effectively alleviate lipid peroxidation in *S. japonica*. The glutathione (GSH) content in the treatment groups showed an increasing trend and was higher than the CK group (Figure 5f). However, there was no significant difference in GSH content among different temperatures in the treatment groups and under 21 °C +ABA conditions.

The Pro content remained stable and relatively low in the CK group (Figure 6a). At all time points, the Pro content was significantly higher in the treated group than in the CK group and slightly decreased after 48 h. At 6 h and 48 h, the difference in Pro content was significant between the groups at 18 °C and 21 °C and between 21 °C and 21 °C +ABA. Compared to the CK group, the Pro content at 48 h increased by 168.51% and 213.7% in the 18 °C and 21 °C groups, respectively, and decreased by 6.76% with the addition of ABA compared to the no-addition variation.

The soluble sugar (SS) content in the treatment group showed a general increasing trend and was significantly higher than CK (Figure 6b). The 21 °C +ABA group was significantly higher than CK and other treatment groups at 6 h. At 12 h, the SS content of the 21 °C group was 31.51% higher than 18 °C. At 24 h, there were no significant differences in the SS content between temperatures and under the 21 °C +ABA condition. The exogenous addition of ABA under high-temperature stress had no significant effect on the SS content of *S. japonica*.

There was no significant change in the soluble protein (SP) content in the CK group (Figure 6c). The trends of the SP content varied greatly among treatment groups. The SP content in the 18 °C group first increased, then decreased, and then increased again, reaching a maximum value of 2.34 mg·g^−1^ Fw at 12 h, which was 78.69% higher than CK. The 21 °C group showed a similar trend to the 18 °C group but reached a maximum at 24 h. The 21 °C +ABA group showed a jagged change and reached a maximum value of 3.12 mg·g^−1^ Fw at 12 h, which increased by 82.46% compared to 21 °C, while the SP content was significantly different from the 21 °C group before 48 h of cultivation.

Samples at 24 h and 72 h time points were subjected to plant hormone metabolome analysis, focusing on the changes in ABA content (Figure 6d). We found that the ABA content of *S. japonica* increased in the high-temperature treated group compared with the CK group, but there was no significant difference between the 18 °C and 21 °C groups. Under high-temperature stress, the exogenous addition of ABA caused a dramatic increase in the ABA content in *S. japonica*, with ABA levels increasing by 8.7 and 7.6-fold at 24 h and 72 h compared to the 21 °C group.

### 2.4. Effect of Abscisic Acid on Gene Expression of S. japonica under High-Temperature Stress

In this study, to gain a deeper understanding of the impact of exogenous ABA on the physiological and biochemical aspects of *S. japonica* under high-temperature stress, *SjEF1α* was used as the reference gene. The expression levels of antioxidant enzyme-related genes *SjSOD* (Cu), *SjSOD* (Fe), *SjPOD*, *SjCAT*, *SjMDA*, *SjGSH*, and *SjAPX* and two key genes for ABA synthesis, *SjAAO3* and *SjNCED*, were analyzed through quantitative PCR (qPCR) (Figure 7).

The results showed that the expression of antioxidant enzyme-related genes was significantly up-regulated in all treated groups compared to the CK group. At 24 h, the expression of the four genes, *SjPOD*, *SjCAT*, *SjMDA*, and *SjGSH*, differed significantly between the 18 °C and 21 °C groups, with the 21 °C group being up-regulated by 10.38%, 11.57%, 14.89%, and 14.41%, respectively, compared to the 18 °C condition. The expression differences between the *SjSOD* (Cu), *SjPOD*, *SjMDA*, *SjGSH*, and *SjAPX* genes in the 21 °C and 21 °C +ABA groups were significant, in which *SjSOD* (Cu), *SjPOD*, and *SjMDA* were up-regulated by 10.04%, 18.56%, and 28.18% in the 21 °C group compared to the 21 °C +ABA group, respectively, while *SjGSH* and *SjAPX* were down-regulated by 15.16% and 81.61%, respectively. At 72 h, compared with the 18 °C group, the 21 °C group up-regulated *SjSOD* (Fe), *SjPOD*, *SjCAT*, *SjMDA*, and *SjAPX* by 6.95%, 13.38%, 25.01%, 9.24%, and 87.85% respectively. *SjPOD*, *SjMDA*, S*jGSH*, and *SjAPX* were down-regulated by 15.95%, 20.83%, 8.7%, and 57.52%, while *SjCAT* was up-regulated by 13.29% in the 21 °C +ABA group compared to the 21 °C group.

*AAO3* and *NCED* have been reported to be two key genes for ABA synthesis [33]. Homologous alignment and gene retrieval revealed the presence of these genes in *S. japonica*. qPCR results showed that the expression levels of *SjAAO3* and *SjNCED* in the treatment group were significantly higher than CK. Among them, compared to the 18 °C and 21 °C groups, there were significant differences in the expression of *SjNCED* at 24 h and 72 h, while *SjAAO3* did not show significant differences. In the 21 °C +ABA group, compared to the 21 °C group, *SjAAO3* was up-regulated by 8.59% and 41.06% at 24 h and 72 h, respectively, and *SjNCED* was up-regulated by 49.13% and 37.69% at 24 h and 72 h, respectively. These observations suggest that exogenous ABA can significantly increase the expression of ABA synthesis-related genes in *S. japonica*.

## 3. Discussion

### 3.1. Effect of ABA on Photosynthetic Characteristics of S. japonica Under High Temperature Stress

Photosynthesis is the main source of material and energy for plants. Plants are prone to photoinhibition under high-temperatures [34,35]. Fv/Fm is an indicator of the photosystem II (PSII) light energy conversion rate [36]. The decrease in Fv/Fm when plants are stressed is an important indicator of photoinhibition [35,37,38,39]. In this study, the Fv/Fm of *S. japonica* under high-temperature stress decreased with an increase in the stress treatment time, and the higher the temperature, the more significant the decrease. After 72 h of high-temperature cultivation at 21 °C, the Fv/Fm decreased by 24.38% compared to CK. Exogenous ABA increased Fv/Fm in *S. japonica* and reduced the physiological damage to *S. japonica*. This indicates that ABA can alleviate the impact of high-temperature stress on the photosynthetic rate of plants, which is consistent with previous observations in *Chlamydomonas reinhardtii* [40] and *Vicia faba* [41].

Photosynthetic pigments mainly include Chl and carotenoids [42]. Chloroplasts are extremely sensitive to high-temperature stress in photosynthesis [43,44]. A number of reports indicated that plants exposed to high-temperature stress show reduced Chl biosynthesis [45]. Our experimental results showed that the Chl content of *S. japonica* under high-temperature stress was significantly lower than CK, resulting in a color change in *S. japonica* tips, which was consistent with the results of previous studies [46]. The reason may be the disruption of the chloroplast membrane structure under high-temperature stress, which inhibits Chl synthesis and accelerates the degradation of synthesized Chl [47]. Under high-temperature stress, a significant increase in Chl a but an insignificant change in Chl b after the addition of exogenous ABA was observed. In plants, Chl a is the main pigment in photosynthesis, and Chl b plays a supporting role by being synthesized by different pathways [48]. Because its primary function is directly related to photosynthetic efficiency, exogenous ABA treatment might preferentially restore Chl a synthesis to maintain the basic function of photosynthesis.

Carotenoids are an important class of secondary metabolites produced during plants growth and development, playing a significant role in the plant’s response to both biotic and abiotic stresses [49]. Several researchers have found that carotenoids play an active role in heat tolerance [50,51]. Our study observed that the carotenoid content of *S. japonica* was increased under high-temperature stress, contradicting the findings of Yi et al. in cabbage [52]. We also found that the changes in carotenoids were not quite the same across species, with violaxanthin decreasing but zeaxanthin, *α*-carotene, and *β*-carotene increasing. It is hypothesized that under high-temperature stress, *S. japonica* increases the synthesis of carotenoids such as *β*-carotene by adjusting metabolic pathways to adapt to the stressful environment. Interestingly, exogenous ABA significantly increased *S. japonica* zeaxanthin at the end of the high-temperature stress test, indicating its potential role in regulating stress tolerance.

### 3.2. Effect of ABA on Antioxidant Enzyme Activity and Gene Expression of S. japonica Under High-Temperature Stress

The massive accumulation of ROS caused by high-temperature stress leads to oxidative damage to the membranes, protein denaturation, and even cell death [53,54]. The main antioxidant enzymes are CAT, SOD, GSH, POD, and APX [55]. Our study found that under high-temperature stress, the SOD and POD activities in *S. japonica* first increased and then decreased, while the CAT content continued to increase, and the APX content increased significantly at the late stage of *S. japonica*, consistent with previous study [31]. This indicates that under high-temperature conditions, the reactive oxygen species content in the *S. japonica* organism increases, the balance of ROS scavenging is disrupted, and the antioxidant enzyme activities are diminished [46]. After the exogenous ABA addition, the trends of its increase and decrease in SOD, POD, CAT, and APX were slowed down to different degrees at the later stage of cultivation, suggesting that ABA plays a role in helping *S. japonica* resist high-temperature stress, alleviating the oxidative stress of *S. japonica*. Intracellular-reduced GSH can eliminate ROS through non-enzymatic reactions, playing a crucial role as an antioxidant substance [56]. In our study, we found that high-temperature stress significantly increased the GSH content in *S. japonica* but did not significantly change with the addition of ABA, which contradicts the findings of Yang et al. in *Camellia oleifera* [57]. This might be due to the fact that ABA treatment did not induce changes in GSH synthesis or glutathione cycling in *S. japonica* [58,59].

MDA is a product of cell membrane lipid peroxidation, is capable of reacting with proteins and damaging DNA structures, and is widely recognized as an important indicator of the extent of lipid peroxidation damage [60,61]. Our study showed that the MDA content of *S. japonica* significantly increased under high-temperature stress compared to CK. This suggests increased oxidative damage to cell membranes at high-temperatures. Exogenous ABA reduced the MDA content by 32.77% at 48 h compared to the no-addition variation, indicating that the antioxidant system in the ABA treatment group effectively reduces ROS, thereby minimizing membrane lipid damage.

SOD, POD, and CAT are typical inducible enzymes, and their transcript levels reflect the degree of plants stress to some extent [62]. In our study, we found that the changes in the expression levels of antioxidant enzymes related to *SjSOD* (Cu), *SjCAT*, *SjMDA*, *SjGSH*, and *SjAPX* were consistent with those of the corresponding enzymes in *S. japonica* under high-temperature stress, whereas the changes in the expression levels of *SjSOD* (Fe) and *SjPOD* were inconsistent. This may be due to the fact that the genes selected for validation were the most highly expressed genes at the transcript level and did not cover all genes. Exogenous ABA had different effects on gene expression levels at different treatment times. At 24 h, the expression of *SjSOD* (Cu), *SjPOD*, and *SjMDA* genes was significantly reduced, and the expression of *SjGSH* and *SjAPX* was increased, whereas, at 72 h, the expression of the genes of *SjPOD*, *SjMDA*, *SjGSH*, and *SjAPX* was significantly reduced. These findings suggest that ABA can regulate the expression of antioxidant enzyme genes in *S. japonica*, and the effects on *S. japonica* under high-temperature stress may be different at different treatment times, perhaps responding to ABA through different signaling pathways [63].

### 3.3. Effect of ABA on Osmotic Regulatory Substances of S. japonica Under High Temperature Stress

Osmoregulatory substances such as Pro, SS, and SP in plant cells help maintain the osmotic balance and stabilize cell turgor under high-temperature stress [64,65]. In this study, the Pro content of *S. japonica* increased under high-temperature stress, indicating that high-temperature stress induces the accumulation of osmoregulatory substances such as Pro and thus regulates the osmotic pressure of the internal environment and maintains the osmotic equilibrium and stability of the cellular structure [66]. The SP content in *S. japonica* at 21 °C increased significantly and reached the maximum at 24 h, which might be the positive response of *S. japonica* to the temperature change, causing some HSPs to be produced in their bodies. The significant increase in SP content after the exogenous addition of ABA suggests that ABA can increase the SP content in plants, and this increase may be related to the role of ABA in regulating the transport, carbon metabolism, and expression of resistance proteins [67]. Meanwhile, we found that the SS content of *S. japonica* surged under high-temperature stress, which was consistent with the results of previous studies on *Caulerpa lentillifera* [68] and *Porphyra yezoensis* [69]. This might be due to the plants adjusting its allocation of energy and metabolites to increase the SS content by reducing sugar consumption and starch hydrolysis in response to a reduced photosynthetic rate [62]. Unfortunately, there was no significant difference after the addition of ABA under high-temperature stress. This suggests that ABA may not directly regulate the SS content but has a positive regulatory effect on the SP content in *S. japonica*, potentially acting in synergy with other regulatory factors.

### 3.4. Effect of ABA on ABA Content and Synthesis-Related Gene Expression of S. japonica Under High-Temperature Stress

ABA plays a key role in regulating plant growth and development, markedly enhancing stress resistance, particularly under conditions of high-temperature. However, there are no relevant reports on how ABA helps *S. japonica* resist adversity. In this study, we observed that the ABA content in *S. japonica* increased under high-temperature stress, indicating that such stress induced a rapid accumulation of ABA in *S. japonica*. This suggests that ABA plays a role in *S. japonica* acclimatization to high-temperatures, consistent with previous findings in *Camellia sinensis* [70]. More importantly, we first found that the exogenous ABA can significantly increase the ABA content in *S. japonica*. Studies have shown that ABA uptake is dependent on specific transporter proteins [71]. In *Arabidopsis thaliana*, several ABA transporters have been identified, such as AtABCG25, AtABCG31, AtABCG30, AtABCG40, AIT1/NRT1.2, DTX50, and NPF5.1 [72,73,74,75]. Homologous proteins of the ABCG family were also present in *S. japonica*, and the expression levels of their related genes were increased with the addition of ABA under high-temperature stress. Therefore, we hypothesized that *S. japonica* can absorb ABA from the culture environment and perform its physiological functions. However, its specific absorbing and transporting mechanism still needs to be further investigated.

There are two pathways for ABA biosynthesis: the C15 direct pathway and the C40 indirect pathway [33]. Higher plants synthesize ABA mainly by the C40 indirect pathway, starting with methylerythritol 4-phosphate pathway (MEP) and passing through *β*-carotene hydroxylase (BCH/CrtZ), zeaxanthin epoxidase (ZEP), violaxanthin de-epoxidase (VDE), 9-cis-epoxycarotenoid dioxygenase (NCED), xanthoxin dehydrogenase (ABA2), and abscisic aldehyde oxidase (AAO3) [76]. It has been reported that ABA is synthesized in *Neopyropia yezoensis* via a pathway that is similar to the carotenoid pathway in higher plants [77]. Through gene calling and transcriptome analysis, we found that the gene encoding the corresponding synthase was also present in *S. japonica*, leading us to hypothesize that *S. japonica* also synthesized ABA via an indirect pathway. NCED is the rate-limiting enzyme, and AAO3 was shown to catalyze the last step in the ABA biosynthesis pathway [78,79]. Our study revealed that the expression levels of *SjAAO3* and *SjNCED* increased under high-temperature stress. The significant activation of the ABA biosynthesis pathway under high-temperature stress suggests that the increased ABA accumulation may regulate the physiological processes of *S. japonica* to enhance stress tolerance. Additionally, the expression of *SjAAO3* and *SjNCED* was found to increase significantly when ABA was introduced. This phenomenon indicates that the introduction of exogenous ABA may facilitate the synthesis of endogenous ABA by stimulating the relevant signaling pathways, enabling the *S. japonica* to perceive alterations in ABA levels and subsequently up-regulate the expression of *SjNCED* and *SjAAO3*. However, the specific mechanism of action still requires further gene function validation. In plants, the ABA signaling pathway plays a crucial role in response to environmental conditions [80]. Under ABA or stress conditions, the ABA core signaling pathway, which consists of PYRABACTIN RESISTANCE1 (PYR1)/PYR1-LIKEs (PYLs)/REGULATORY COMPONENTS OF ABA RECEPTOR (RCARs), clade A protein phosphatase type 2Cs (PP2Cs), and SNF1-related protein kinase 2s (SnRK2s), is activated to phosphorylate downstream components, such as transcription factors ABI3 and ABI5 [81,82]. Transcriptome analysis has identified the presence of ABIs, PP2Cs, and SnRK2s in *S. japonica*, providing a foundation for the further exploration of the specific mechanisms by which exogenous abscisic acid influences *S. japonica* in future studies.

## 4. Materials and Methods

### 4.1. Experimental Materials

Seawater for the experiment was collected offshore in Xiamen, Fujian Province, China, and sterilized prior to use. *S. japonica* sporophytes (20–30 cm) were collected from Xiapu, Fujian Province, China (119°92′ E, 26°56′ N), in mid-November 2023, washed several times with sterilized low-temperature seawater and temporarily cultured in algae incubator for 3 days. Cultivation conditions: 12 °C, LED white light, light intensity of 50 μmol·m^−2^·s^−1^, and photoperiod of 12L:12D. A culture medium was sterilized with natural seawater, and the nutrient salts were a NaNO_3_ solution as the nitrogen source and KH_2_PO_4_ as the phosphorus source, with the concentrations of NO_3_^−^-N at 4 mg/L and PO_4_^−^-P at 0.4 mg/L. An aerated suspension culture with water was changed every 3 days.

### 4.2. Experimental Design

The experiment consisted of four groups: a normal culture group (CK), a high-temperature treatment group at 18 °C (18 °C), a high-temperature treatment group at 21 °C (21 °C), and an exogenous addition of 10 μM ABA to the 21 °C high-temperature treatment group (21 °C +ABA). Each group had 4 replicates, and samples were collected after 6, 12, 24, 48, and 72 h for analysis.

### 4.3. Determination of Morphology

Prior to sampling at each sampling point, the macroscopic specimen imaging system Longbase DT810 (Qingdao Longbase Medical Device Co., Ltd., Qingdao, China) was used to record the growth status of *S. japonica* under high-temperature stress and the effect of exogenous ABA on the growth status of *S. japonica* under high-temperature stress.

### 4.4. Growth and Survival Measurement

The RGR and survival rate were calculated by weighing the fresh mass at the beginning and at the end of the incubation and counting the number of survivors in the culture system. The relative growth rate was calculated as follows: RGR = [Ln (Wt/W0)/t] × 100%, where W0 is the initial fresh weight of algae (g), Wt is the fresh weight of algae at the end of the experiment (g), and t is the duration of the experiment (d). The survival rate was calculated as follows: survival rate = number of survivors/total number.

### 4.5. Determination of Chlorophyll Fluorescence Parameters and Chlorophyll Content

Chlorophyll fluorescence parameters were determined using the DUAL-PAM-100 (WALZ, Bavaria, Germany). *S. japonica* was dark-adapted for 20 min before the Fv/Fm was measured. Chl content was determined using the Plant Chlorophyll Content Assay Kit (Solarbio, Beijing, China). The total Chl content and Chl a and Chl b content were measured.

### 4.6. Determination of Physiological and Biochemical Indices

In this study, SOD, POD, CAT, and APX activities, as well as MDA, GSH, Pro, and SS contents, were determined using kits purchased from Beijing Solarbio Science & Technology Co., Ltd. (Solarbio, Beijing, China). The content of SP was determined using kits purchased from Ruixin Bio-Tech Co., Ltd. (Ruixinbio, Quanzhou, China).

### 4.7. Determination of ABA Content

Sample extraction: approximately 100 mg of frozen material samples were extracted in 1 mL of ice-cold 50% aqueous Acetonitrile (ACN, *v*/*v*). The samples were sonicated for 3 min at 4 °C and subsequently extracted using a benchtop laboratory rotator for 30 min at 4 °C. After centrifugation (10 min, 12,000 rpm, 4 °C), the supernatant was transferred to clean plastic microtubes. All samples were purified using C18 reversed-phase, polymer-based, solid-phase extraction (RP-SPE) cartridges that had been washed with 1 mL of methanol (MeOH) and 1 mL of deionized water and then equilibrated with 50% aqueous ACN (*v*/*v*). After loading a sample, the cartridge was then rinsed with 1 mL of 30% ACN (*v*/*v*), and this fraction was collected. After this single-step SPE, the samples were evaporated to dryness under a gentle stream of nitrogen and stored at −20 °C until analysis. For UHPLC-ESI-MS/MS analysis, the samples were dissolved in 200 μL of 30% ACN (*v*/*v*) and transferred to insert-equipped vials.

UPLC conditions: The sample extracts were analyzed using a UPLC-Orbitrap-MS system (UPLC, Vanquish, Thermo Fisher Scientific, Waltham, MA, USA; MS, QE). The analytical conditions were as follows, UPLC: column, Waters ACQUITY UPLC HSS T3(1.8 μm, 2.1 mm × 50 mm); column temperature, 40 °C; flow rate, 0.3 mL/min; injection volume, 2 μL; solvent system, water (0.1% acetic acid): acetonitrile (0.1% acetic acid); gradient program, 85:15 *v*/*v* at 0 min, 85:15 *v*/*v* at 0.5 min, 10:90 *v*/*v* at 1.5 min, 10:90 *v*/*v* at 3 min, 85:15 *v*/*v* at 3.1 min, and 85:15 *v*/*v* at 5.0 min.

LC-MS/MS analysis: HRMS data were recorded on a Q Exactive hybrid Q-Orbitrap mass spectrometer equipped with a heated ESI source (Thermo Fisher Scientific, Waltham, MA, USA) utilizing the SIM MS acquisition methods. The ESI source parameters were set as follows: spray voltage, −2.8 kV; sheath gas pressure, 40 arb; aux gas pressure, 10 arb; capillary temperature, 320 °C; and aux gas heater temperature, 350 °C.

### 4.8. Determination of Carotenoids

Sample extraction: 0.1–0.5 g well-homogenized pooled sample was placed in a 15 mL tube with 2 mL absolute ethanol and 0.1% Butylated hydroxytoluene (BHT). After vortex mixing, the tube was placed in an 80 °C water bath for 5 min. After removal from the water bath, 100 μL of potassium hydroxide solution (80% *w*/*v*, in water) was added. The sample was vortexed and then returned to the water bath for 15 min. The tube was added with 1 mL water and then added with 1 mL n-hexane. After being centrifuged at 3000 rpm for 5 min, the supernatant was removed and put into a fresh test tube, and the residue was re-extracted with 1 mL n-hexane. The test tube was centrifuged, and the supernatant was combined with the above extract. This procedure was repeated one more time using 1 mL n-hexane, followed by the evaporation to dryness under a gentle flow of nitrogen at 30 °C. The residue was reconstituted with 0.2 mL of methanolic solution.

UPLC conditions: The sample extracts were analyzed at 450 nm using a UPLC system with a DAD detector (UPLC, U3000, Thermo Fisher Scientific, Waltham, MA, USA). The analytical conditions were as follows, UPLC: column, YMC Carotenoid S-3 μm (150 × 4.6 mm); column temperature, 40 °C; flow rate, 1.0 mL/min; injection volume, 2 μL; solvent system, MeOH: (MeOH:MTBE:H_2_O = 20:75:5); gradient program, 100:0 *v*/*v* at 0 min, 39:61 *v*/*v* at 15 min, 0:100 *v*/*v* at 25 min, 100:0 *v*/*v* at 25.1 min, and 100:0 *v*/*v* at 30 min.

Data analysis: Data were acquired on the U3000 UPLC (Thermo Fisher Scientific, Waltham, MA, USA) and processed using chromeleon 7.2 CDS (Thermo Fisher Scientific, Waltham, MA, USA). Quantified data were output into an Excel format.

### 4.9. RNA Extraction and Real-Time Fluorescent Quantitative Analysis

Total RNA from *S. japonica* was extracted using the HP Total RNA Kit (Omega Bio-tek, Norcross, GA, USA), and the first strand cDNA was synthesized using the TransScript^®^ All-in-One First-Strand cDNA Synthesis SuperMix for qPCR (One-Step gDNA Removal) (TransGen Biotech, Beijing, China). The cDNA was then diluted three times to serve as a template for real-time fluorescent quantitative PCR (qRT-PCR), with *EF1α* as the reference gene. The qRT-PCR reagents used were PerfectStart^®^ Green qPCR SuperMix (TransGen Biotech, Beijing, China). Based on the transcriptome data under *S. japonica* stress conditions, the sequences of the genes with the most pronounced up-regulation of expression in each gene were screened for transcriptional studies, and the nucleic acid sequences of the target molecules were obtained from the *S. japonica* genome data. Primers required for the qRT-PCR experiment were designed using Primer Premier 5 software (Premier Biosoft, Palo Alto, CA, USA), and their sequences are shown in Table 1. The 2^−∆∆Ct^ method was used to calculate the relative transcription levels of the genes.

### 4.10. Data Analysis

In this study, data collection and preliminary processing were conducted using Microsoft Excel 2021 software (Microsoft Corporation, Redmond, WA, USA). The statistical analysis of the experimental data was performed using IBM SPSS Statistics 27 software (IBM Corporation, Armonk, NY, USA). To assess the significant differences in photosynthetic parameters and physiological indicators among various treatment groups, an analysis of variance (ANOVA) was initially conducted, followed by Duncan’s multiple range test to elucidate the disparities between groups. In the presentation of results, distinct letters indicate significant differences between treatment groups at the same time point. A *p*-value of less than 0.05 was considered statistically significant. Graphs and charts were generated using GraphPad Prism 9 software (GraphPad Software, San Diego, CA, USA).

## 5. Conclusions

In this study, we investigated various physiological and biochemical changes (including ABA) in *S. japonica* under high-temperature stress and the physiological and molecular regulatory mechanisms of exogenous ABA on *S. japonica* resistance to adversity. It was found that under high-temperature stress (21 °C), *S. japonica* exhibited severe rot, decreased photosynthesis, increased related antioxidant enzymes and osmoregulatory substances, and elevated transcript levels of genes related to antioxidant enzymes. Further studies revealed that the treatment of exogenous ABA significantly ameliorated the damage caused by high temperatures, regulated the physiological state of *S. japonica* through multiple pathways such as increased photosynthesis, reduced oxidative stress, and promoted the synthesis of osmoregulatory substances. It also altered the transcript levels of antioxidant enzyme-related genes, mitigating the oxidative damage caused by high-temperature stress. Additionally, content assays and gene expression analyses showed that *S. japonica* could absorb exogenous ABA and regulate the expression of related genes, which increased the sensitivity of *S. japonica* to the ABA metabolism pathway, improved the stability of membrane lipids and enhanced the resilience. In conclusion, this study provides evidence that exogenous ABA can moderate the deleterious effects of high-temperature stress, provides a new solution to the losses of *S. japonica* due to high temperature in actual production culture, and provides a theoretical basis for verifying the function of *S. japonica* genes, screening germplasm resources, and breeding new *S. japonica* cultivars resistant to high-temperature disease and decay.

## Figures and Tables

**Figure 1 ijms-25-11581-f001:**
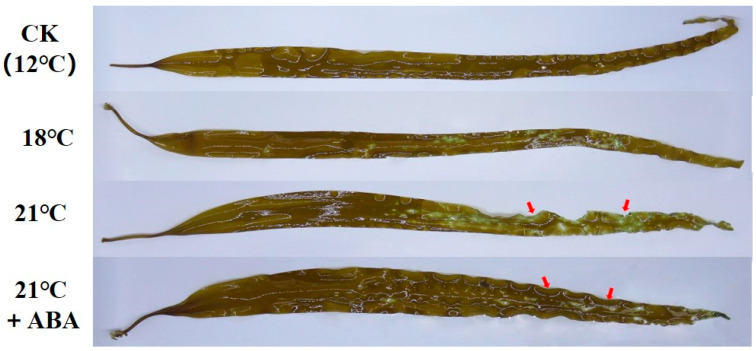
Effect of ABA on physiological morphology of *S. japonica* under high-temperature stress. The red arrows highlight the most significant portions of the change.

**Figure 2 ijms-25-11581-f002:**
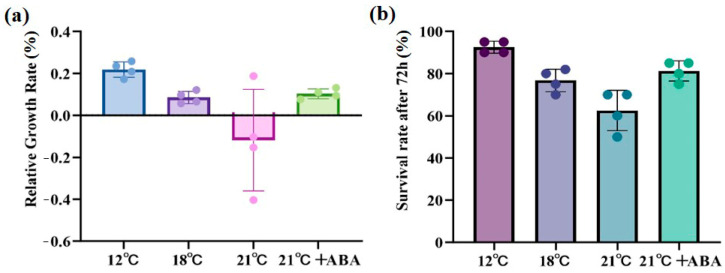
Effect of ABA on physiological characterization of *S. japonica* under high-temperature stress. (**a**) Relative growth rate, (**b**) Survival rate. Data are expressed as means ± standard deviations (SD). The colored dots represent data points for *n* = 4.

**Figure 3 ijms-25-11581-f003:**
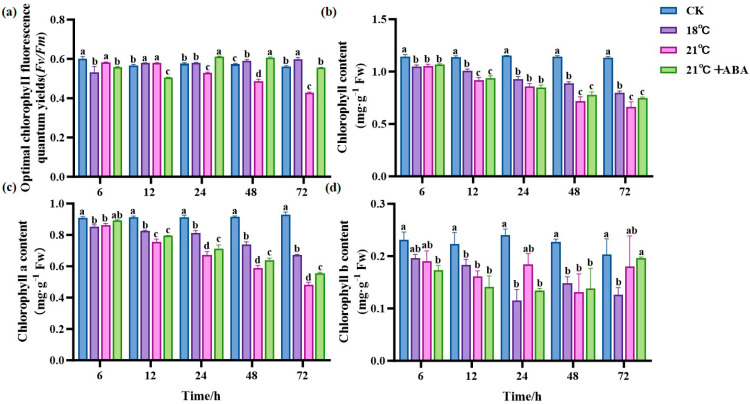
Effect of ABA on photosynthesis in *S. japonica* under high-temperature stress. (**a**) Fv/Fm, (**b**) Total Chl content, (**c**) Chl a content, (**d**) Chl b content. Data are expressed as means ± standard deviations (SD). According to Duncan’s multiple range test, different letters indicate significant differences between the means of different treatments at the same time point (*p* < 0.05).

**Figure 4 ijms-25-11581-f004:**
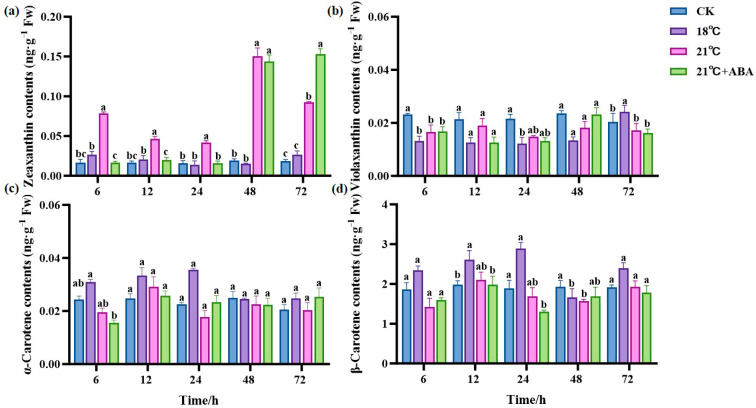
Effect of ABA on carotenoid content of *S. japonica* under high-temperature stress. (**a**) Zeaxanthin, (**b**) Violaxanthin, (**c**) *α*-carotene, (**d**) *β*-carotene. Data are expressed as means ± standard deviations (SD). According to Duncan’s multiple range test, different letters indicate significant differences between the means of different treatments at the same time point (*p* < 0.05).

**Figure 5 ijms-25-11581-f005:**
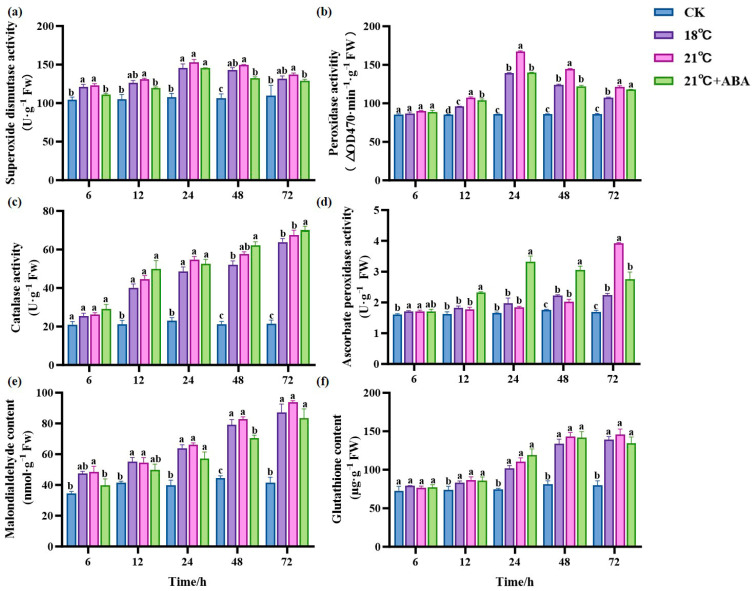
Effect of ABA on the antioxidant system of *S. japonica* under high-temperature stress. Including SOD (**a**), POD (**b**), CAT (**c**), APX (**d**), MDA (**e**) and GSH (**f**). Data are expressed as means ± standard deviations (SD). According to Duncan’s multiple range test, different letters indicate significant differences between the means of different treatments at the same time point (*p* < 0.05).

**Figure 6 ijms-25-11581-f006:**
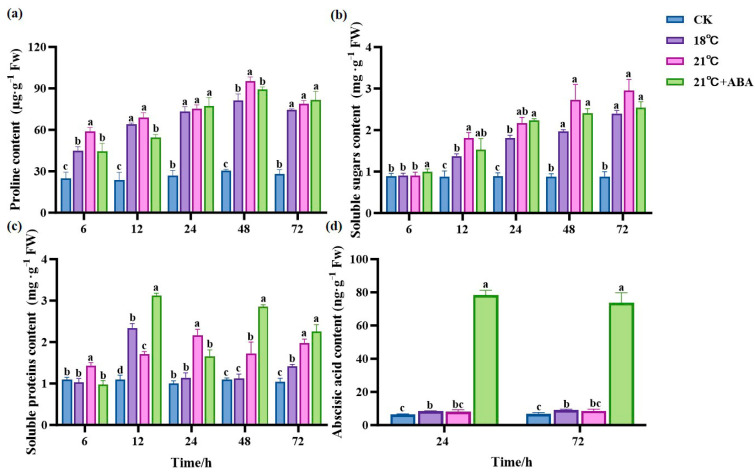
Effects of ABA on the content of Pro (**a**), SS (**b**), SP (**c**) and ABA (**d**) in *S. japonica* under high-temperature stress. Data are expressed as means ± standard deviations (SD). According to Duncan’s multiple range test, different letters indicate significant differences between the means of different treatments at the same time point (*p* < 0.05).

**Figure 7 ijms-25-11581-f007:**
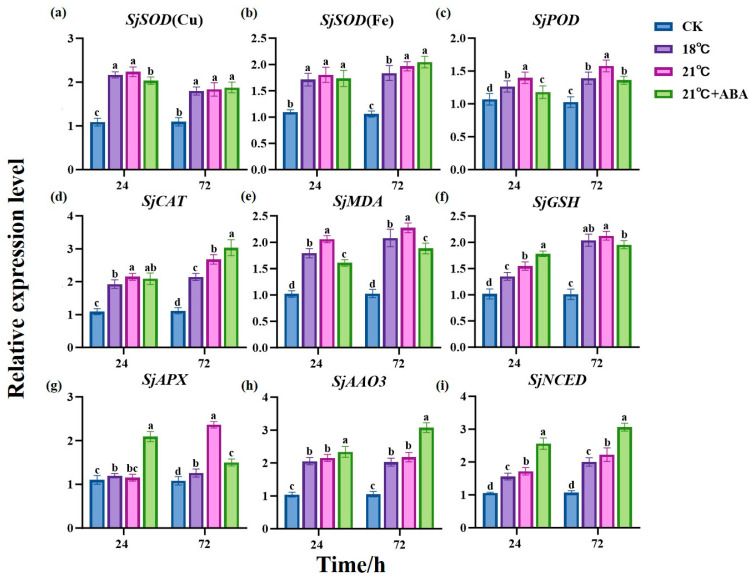
Effects of ABA on the expression of antioxidant enzyme-related genes *SjSOD* (Cu) (**a**), *SjSOD* (Fe) (**b**), *SjPOD* (**c**), *SjCAT* (**d**), *SjMDA* (**e**), *SjGSH* (**f**), *SjAPX* (**g**) and two key genes for ABA synthesis, *SjAAO3* (**h**) and *SjNCED* (**i**), in *S. japonica* under high-temperature stress. Data are expressed as means ± standard deviations (SD). According to Duncan’s multiple range test, different letters indicate significant differences between the means of different treatments at the same time point (*p* < 0.05).

**Table 1 ijms-25-11581-t001:** Gene primer sequences used for the quantitative real-time PCR analysis.

Gene	Forward Primer (5′-3′)	Reverse Primer (5′-3′)
*SjEF1α*	GTGATGGAGGAGAACCC	TTGATGACACCCACAGC
*SjSOD* (Cu)	GACTTGTCGGAGGGTTTGGT	CGTCGATGTTCCCCAAGTCT
*SjSOD* (Fe)	CCATCGAGAAGGAGTACGGC	CCTCCTTGATGGGGTTGGAC
*SjPOD*	TCGGAGATGAGGGGATCGTT	GCTGTTGTCGAACTTGAGCC
*SjCAT*	CAACCCCTTCGATGTCACCA	TCCAGGAACCATGTTGGACG
*SjMDA*	ACACACACACGGGAACCTAC	CCATTATTACCGCCTCCGCT
*SjGSH*	CCTCTTTCGGGTGCATGAGT	CCGTGAGGGTCGGATTATCG
*SjAPX*	CCAAGGTGTACAGACGGGAC	GCAGAACAAATGCTGCGGAA
*SjAAO3*	CAACATGTACAAGGAGGGCG	CTTGGTCGGAATGACCGAAAG
*SjNCED*	GGCGAATGCGTGTTCATACC	GCTTCGCGTTCATGGTCTTC

## Data Availability

Data are available on request from the corresponding authors.

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
