# Peer review of "Effects of Abscisic Acid on the Physiological and Biochemical Responses of *Saccharina japonica* Under High-Temperature Stress"

_ijms, 2024, doi:10.3390/ijms252111581_

Round 1

Reviewer 1 Report

Comments and Suggestions for Authors

This work is aimed to evaluate the role and effect of abscisic acid (ABA) on the stress-regulating mechanisms of the aquatic plant Saccharina japonica.  The topic is interesting and fits into the Journal’s scope: the work is well written, the results are adequately discussed, and the conclusions are well supported by data. I only found some minor flaws and typos, which are reported in the following lines.

Line 97: Please use the extended form for all abbreviated names, when they are used for the first time. This is for RGR (line 97), Fv/Fm(line 124), Chl (line 129) and elsewhere in the text.

Caption to figure 1: the figures should stand alone. In (a) explain what the red arrows indicate; in (b) and (c) explain the meaning for the symbols reported: are the bars on top of the histograms SD or SE? And what are the colored dots for? In the Y axis, the measurement units should be also indicated. The meaning of the bars on histograms should be added also in all other figures.

Line 177: check the verb in the sentence: “..all treatment groups was higher…”

Line 281: substitute “plant” with “plants”

Lines 297-299: a verb seems to be missing, please check the sentence.

Line 357: substitute “difference” with “different”

Line 438: here, the verb “use” seems in the imperative form; modify the sentence using the past tense, as in the other parts of the M&M section.

Line 455: substitute “can be measured” with “were measured”.

Figure 7: the inclusion of a figure in the “conclusion” section, although possible, is not usual in this Journal. Check the opportunity to move it as a graphical abstract (as it actually is). As reported in the Instruction for Authors, in addition to summarizing the content, it could “represent the topic of the article in an attention-grabbing way”.

Comments on the Quality of English Language

English form is fine. Some minor typos and flaws are listed in the review report.

Reviewer 2 Report

Comments and Suggestions for Authors

The manuscript “ Effects of Abscisic Acid on the Physiological and Biochemical Responses of Saccharina japonica under High-Temperature Stress” was reviewed carefully and found interesting. The manuscript explains the effect of the exogenous application of ABA on high-temperature stress tolerance in S. japonica.

The authors analyzed relative growth rate, chlorophyll fluorescence parameters (Fv/Fm), photosynthetic pigment, and osmotic substance content. In addition to the physiological studies, the authors explain the molecular mechanism by studying the expression of ABA synthesis-related genes. This study may be further useful for screening S. japonica germplasm resources under stress conditions. 

The manuscript may be accepted with the following minor corrections:

The statistical significance “p value should be in italics.

Include the accession numbers for genes listed in Table 1; this will be useful for the readers.

A few of the important latest references were missing in the discussion, please cite.

The present manuscript is written well to address the focused objective i.e. The effect of ABA on the elevation of high-temperature stress on Saccharina japonica.

Comments on the Quality of English Language

 Minor editing of English language required
